

# Random forests with parametric entropy-based information gains for classification and regression problems

Vera Ignatenko,  Anton Surkov and  Sergei Koltcov

Social and Cognitive Informatics Laboratory, National Research University Higher School of Economics, Saint-Petersburg, Russia

## ABSTRACT

The random forest algorithm is one of the most popular and commonly used algorithms for classification and regression tasks. It combines the output of multiple decision trees to form a single result. Random forest algorithms demonstrate the highest accuracy on tabular data compared to other algorithms in various applications. However, random forests and, more precisely, decision trees, are usually built with the application of classic Shannon entropy. In this article, we consider the potential of deformed entropies, which are successfully used in the field of complex systems, to increase the prediction accuracy of random forest algorithms. We develop and introduce the information gains based on Renyi, Tsallis, and Sharma-Mittal entropies for classification and regression random forests. We test the proposed algorithm modifications on six benchmark datasets: three for classification and three for regression problems. For classification problems, the application of Renyi entropy allows us to improve the random forest prediction accuracy by 19–96% in dependence on the dataset, Tsallis entropy improves the accuracy by 20–98%, and Sharma-Mittal entropy improves accuracy by 22–111% compared to the classical algorithm. For regression problems, the application of deformed entropies improves the prediction by 2–23% in terms of R2 in dependence on the dataset.

## INTRODUCTION

The world has experienced an increase in the amount and variety of available data, which needs to be analyzed. This has given rise to universal algorithms that may, in some sense, select data attributes, discard background noise, and give the analyst a comprehensible summary for understanding complex datasets. To better address this need, a decision tree algorithm was proposed. One of the earliest works on decision trees is the book by *Breiman (1984)*, where the authors describe the basic concepts and algorithms of decision trees and their application for classification and regression problems. In the 1990s, researchers found that using ensembles produced better prediction accuracy than single weak learners. One of the earliest works on ensemble methods is the boosting algorithm (*Schapire, 1990*), which discusses how iterative re-weighting of training data can be used to construct a strong classifier as a linear combination of a large number of weak classifiers. The

Corresponding author
Vera Ignatenko,
veraignatenko93@gmail.com

combination of the decision tree concept and ensemble methods led to the emergence of a random forest, that is, an ensemble of trained decision trees. Breiman's works (*Breiman, 1996*; *Breiman, 2001*) popularized the use of the random forest. In these works, the author introduced bagging into the random forest algorithm, *i.e.,* random sampling of training data, which, in turn, improved the prediction accuracy of random forest. Random forest algorithms have become widespread due to their simplicity and relatively high prediction accuracy in a broad class of problems. In addition, these algorithms have a small number of parameters and are easily parallelizable (*Biau & Scornet, 2016*). For example, the random forest methodology has been successfully used in molecular informatics (*Svetnik et al., 2003*), ecology (*Prasad, Iverson & Liaw, 2006*; *Cutler et al., 2007*), hydrology (*Tyralis, Papacharalampous & Langousis, 2019*), medicine (*Sarica, Cerasa & Quattrone, 2017*), genetics (*Chen & Ishwaran, 2012*), 3D object recognition (*Shotton et al., 2013*), bioinformatics (*Díaz-Uriarte & de Andrés, 2006*; *Qi, 2012*), and for text tone analysis (sentiment analysis) (*Stephenie, Warsito, Budi & Prahutama, Alan, 2020*; *Karthika, Murugeswari & Manoranjithem, 2019*). In addition, in *Howard & Bowles (2012)*, the authors postulate that ensembles of decision trees, *i.e.,* random forests, are the most successful universal and multipurpose algorithms of our time. *Grinsztajn, Oyallon & Varoquaux (2022)* demonstrate that random forest algorithms outperform neural networks on tabular data.

Despite the popularity and broad application of random forests, we would like to note that these algorithms are based on the use of information gain with classical Shannon entropy (*Breiman, 2001*; *Criminisi, Shotton & Konukoglu, 2012*). However, it is known that complex data are better described with nonclassical deformed entropies. Such data emerge in different fields of knowledge, for instance, in physical and biological systems, as well as in language models and the behavior of financial markets. For example, *Tsallis (2009)* described a general approach to modification of mathematical formalism for describing non-extensive physical systems based on a combination of different entropies. *Beck (2009)* presented a general overview of the most frequently used parameterized entropies in the scientific literature. *Clauset, Shalizi & Newman (2009)* proposed the mathematical formalism for describing power-law distributions in data of different types (distribution of words in English-language texts, the degrees of proteins in the partially known protein-interaction network, the number of species per genus of mammals and others). Let us note that this type of distribution naturally follows from Renyi entropy (*Bashkirov & Vityazev, 2007*). *Bohorquez et al. (2009)* considered examples of complex data generated by different human activities, including the behavior of financial markets. For instance, *Li, He & Song (2016)* considered the behavior of the Chinese market based on three types of entropy: Shannon, Renyi, and Tsallis entropy. Thus, parameterized entropies are actively used in the analysis of different types of data.

Recent studies by *Gajowniczek, Zabkowski & Orłowski (2015)*, and *Maszczyk & Duch (2006)* show that the application of nonclassical parameterized entropy can lead to promising results and increase the prediction accuracy of decision trees. To the best of our knowledge, one-parametric Renyi and Tsallis entropies were applied only for the classification tasks. Moreover, deformed entropies were used and tested only for

constructing individual decision trees, but not random forests. Thus, this work aims to partially fill this gap and test nonclassical entropies' application to both classification and regression tasks in random forest algorithms. Moreover, we considered one-parametric entropies such as Renyi and Tsallis as well as two-parametric Sharma-Mittal entropy. For the classification tasks, we estimated the results in terms of accuracy and F1 scores. For the regression tasks, we calculated mean squared error, mean absolute error, and coefficient of determination to evaluate the effect of applying nonclassical entropies compared to the classical algorithm (namely, Breiman's random forest algorithm *Breiman, 2001*). We conducted experiments on three datasets for the classification task and three datasets for the regression task. The datasets originate from different fields (such as human activity recognition, brain activity, physics, finances, and health) and have different sizes.

Here, we proposed a modification of the random forest algorithm, in which the splitting procedure is modified based on the application of parametrized entropies such as Renyi, Tsallis, and Sharma–Mitall entropy. This approach is a generalization of the random forest algorithm since Shannon entropy is a special case of Renyi and Tsallis entropies, which, in turn, are special cases of Sharma-Mittal entropy. Parameterized entropies allow us to add additional parameters to effectively configure the random forest algorithm for various patterns in complex data, and our experiments demonstrate that the introduction of parameterized entropies significantly improves the quality of classification in classification problems and the quality of prediction in regression problems on various datasets. More precisely, the new algorithm's performance in classification problems is better by at least 22% in terms of accuracy and by at least 3% in terms of $R^2$ compared to the classical random forest algorithms.

The rest of our work consists of the following parts. The 'Basics of random forests' section describes the fundamental rules of random forest construction for classification and regression tasks. The 'Application of parametric entropies in random forests' section describes the proposed variations of the target function, namely, different types of information gain based on different types of parametric entropy, for finding the best split in the tree construction process and general algorithm of random forest construction. The 'Description of computer experiments' section outlines the design of our computer experiments for each type of proposed information gain and describes the used datasets. The 'Results' section demonstrates the behavior of quality measures in dependence on the type of information gain and corresponding parameters and outlines the best results for each dataset. The 'Discussion' section interprets the obtained results and reviews the possible limitations of the proposed approach. The 'Conclusion' section summarizes our findings.

## BASICS OF RANDOM FORESTS

The basic idea of a random forest (an ensemble of decision trees) is to create hierarchically tree-like structures consisting of decisive rules of the form "If ... then ...". The rules are automatically generated during the learning process on the training set. With this approach, the dataset is divided into many pieces, and its own set of rules is formed for each piece.

The formation of the final set of rules is determined in such a way that tree branching (splitting of data) leads to a decrease in Gini impurity or Shannon entropy in the case of classification task or to minimization of the error for the regression model. The result of learning a random forest is a set of decision rules. Accordingly, a set of such rules is used to predict a class (classification model) or a real number (regression model). In the framework of this work, we propose to use the parametrized entropies of Renyi, Tsallis and Sharma-Mittal instead of the classical version of the Shannon entropy. Correspondingly, the use of additional entropy parameters in the formation of splitting rules makes it possible to increase the quality of the random forest algorithm.

Let us consider the mathematical formalism of constructing such rules in more detail. We begin by describing the concept of a decision tree. A tree is a data structure in the form of a set of linked nodes represented as a connected graph without loops. Nodes can be one of two types: internal and leaf (terminal) nodes. Each node has only one input edge. In this work, we will consider only binary trees, where each internal node has exactly two edges exiting from it. A decision tree is a set of questions organized hierarchically and represented graphically as a tree. For a given input object, the decision tree evaluates an unknown property of the object using a sequence of questions about known object properties. Each internal node is associated with one such question. The sequence of questions is represented graphically as a tree path. A decision is then made based on the leaf node in the path. To build a decision tree, the following three rules are required (*Breiman, 1984*):

1. The way to choose the splitting of each internal node (that is, setting the target function to be optimized by various possible splits).
2. The rule that determines that the node is terminal.
3. Prediction rule for each terminal node (*i.e.,* a class for the classification problem and a number in the case of regression).

The training phase of the decision tree consists of optimizing the selected target function defined on the available training dataset. The optimization takes place in the form of a greedy algorithm. Let us denote by $S_j$ the set of data points falling into node $j$. Then, the target function $I_j$ is optimized by iterating through the possible splits for each node $j$, where $I_j = I(S_j, S_j^L, S_j^R)$, and $S_j^L, S_j^R$ are data points falling into right and left subtrees respectively. The type of the target function depends on the problem.

Let us discuss the first rule in more detail. For the classification problem, information gain (*Criminisi, Shotton & Konukoglu, 2012*) or Gini Index (Gini impurity) (*Breiman, 1984*) is usually used as the target function. Information gain is based on the concept of entropy, which in turn, is a measure of uncertainty or disorder. Information gain is used to quantify which feature provides the most information about the classification based on the concept of entropy, *i.e.,* it measures the quantity of uncertainty in order to reduce the amount of entropy going from the top (root node) downwards (terminal nodes). Information gain is defined as follows:

$$I_j = H(S_j) - \sum_{i \in \{L,R\}} \frac{|S_j^i|}{|S_j|} H(S_j^i), \tag{1}$$

where $H(\cdot)$ is Shannon entropy, *i.e.,* $H(S) = -\sum_{c \in C} p(c) \log p(c)$, $C$ is the set of classes, $c$ is a class label. It is worth noting that in *Maszczyk & Duch (2006)*, and *Gajowniczek, Zabkowski & Orłowski (2015)*, nonclassical entropies were considered for decision tree construction instead of classical Shannon entropy. Specifically, Renyi and Tsallis entropies are applied in these works. However, numerical experiments were performed only for the binary classification problem and only for the C4.5 algorithm (*Quinlan, 1993*). The results of these works show that the application of nonclassical entropies for constructing decision trees is a promising direction.

In contrast to information gain, Gini Index estimates the probability of misclassification for a randomly selected object from a given node. If all elements in a given node belong to the same class, then such a node can be called "pure". The features with the lowest Gini Index value are chosen during the tree construction. Mathematically, Gini Index is defined as follows: $Gini = 1 - \sum_{c \in C} p^2(c)$.

For the regression problem, the least-square mean or mean absolute error is usually used as the target function (*Breiman, 1984*). The target function in the form of the least-square mean in the case of a one-dimensional dependent variable is expressed as follows:

$$\delta R = R(S_j) - R(S_j^L) - R(S_j^R), \tag{2}$$

where $R(S_j) = \frac{1}{|S_j|} \sum_{x_i \in S_j} (y_i - \bar{y}_j)^2$, $\bar{y}_j = \frac{1}{|S_j|} \sum_{x_i \in S_j} y_i$, *i.e.,* $\bar{y}_j$ is the sample mean of values falling into the given node. Thus, $R(S_j)$ is the sum of the squares of the deviations of the values falling in a given node from the average value for that node. The tree is constructed by iterative splitting in such a way as to maximize the target function and correspondingly maximize the reduction of $R(S)$.

The target function in the form of mean absolute error is defined as follows (*Breiman, 1984*): $\delta R = R(S_j) - R(S_j^L) - R(S_j^R)$, where $R(S_j) = \frac{1}{N} \sum_{x_i \in S_j} |y_i - \hat{v}_j|$, $\hat{v}_j$ is the sample median of values $y_i$ falling in the node $j$.

Let us note that in *Criminisi, Shotton & Konukoglu (2012)*, the target function based on information gain using the classical Shannon entropy is proposed for the construction of the regression tree: $I_j = \sum_{x \in S_j} \log(|\Lambda_y(x)|) - \frac{|S_j^i|}{|S_j|} \sum_{i \in \{L,R\}} (\sum_{x \in S_j^i} \log(|\Lambda_y(x)|))$, where $\Lambda_y$ is the conditional covariance matrix. In *Nowozin (2012)*, the application of information theory and the use of information gain based on Shannon entropy to construct a regression tree are also discussed.

The second rule for constructing a decision tree in the framework of random forest is usually formulated as follows: a node becomes terminal either when the tree reaches the maximum allowable number of levels $D$, or the maximum of the target function becomes less than some given minimum value, or the node contains too little training data, that is, less than some predetermined number.

Let us discuss the third rule for constructing a decision tree, which also depends on the problem type. For the classification problem, the class to which the largest number of training data corresponds in the given leaf node serves as the predictor (majority principle) (*Breiman, 1984*). Moreover, instead of predicting one class, we can obtain the probability of each class $p(c|x), c \in C$, which is equal to the ratio of the number of training

data with a given class in the given leaf node to the total number of training data in the given leaf node. For the regression problem, a predictor in the form of a constant is usually used (*Breiman, 1984*); namely, the sample mean $\bar{y}_j$ or the sample median $\hat{v}_j$ of all training data that fell into a given node. If a sample mean is chosen as the predictor, then the least-square error is considered the target function. If a sample median is chosen as a predictor, then the mean absolute error is taken as the target function. In *Criminisi, Shotton & Konukoglu (2012)*, a probability-linear function is considered a predictor, namely, $y = \beta_0 + \beta_1 x_1 + \ldots + \beta_p x_p$, and, thus, each leaf node produces a conditional probability $p(y|x)$. In this case, information gain based on Shannon differential entropy is used as the target function.

Next, let us discuss the details of the random forest construction. A random forest is an ensemble of randomly trained decision trees. The key aspect of the forest model is that its components, *i.e.*, decision trees, differ from each other in a random manner. Each tree is trained on a random subset of the original training data, and then the individual predictions from each tree are aggregated to form the final prediction.

In this case, randomness can be introduced in various ways. First, random subsets of observations, rather than all available observations, can be used in the construction of each tree. If sampling without repetition is performed, such a method is called "pasting" (*Breiman, 1999*). If sampling is performed with repetition, then this method is called bagging (*Breiman, 1996*; *Breiman, 2001*). Let us describe bagging in more detail since it is almost always used in practical applications. If there are $n$ observations in the training dataset $W$, then bootstrap samples $W_k$ of size $n$ are formed in the bagging process, and then each tree is built based on one such bootstrap sample. Thus, in each bootstrap sample, some observations will be selected more than once, and some observations will not be selected at all. It can be shown that each bootstrap sample will be missing approximately 1/3 of the observations from the original dataset. Since each tree is constructed using only about 2/3 of the data, most of the trees will be significantly different from each other. In addition, using bootstrap samples produces out-of-bag (OOB) estimates. OOB samples are about 1/3 of the observations that were not used to construct a particular tree. Thus, one can test each $x$ that has not been used for the particular tree construction and calculate the average prediction error for such values. The OOB errors are then calculated for each tree, and then these values are averaged to estimate the accuracy of the predictions of the random forest. The estimate constructed in this way is essentially leave-one-out cross-validation. Bagging allows us to get an estimate of model accuracy without formally testing on new data.

Second, random subspaces of a dataset can be obtained by randomly sampling attributes (features). In practice, the number of attributes $m = \sqrt{p}$ is often chosen, where $p$ is the total number of attributes (*Probst, Wright & Boulesteix, 2019*). This method is called "random feature selection" (*Breiman, 2001*) or "random subspaces" (*Ho, 1998*).

Third, each tree can be constructed simultaneously on a subset of observations and a subset of attributes of observations. This method is called "random patches" (*Louppe & Geurts, 2012*).

The introduction of randomization leads to the decorrelation of predictions between trees. Thus, the predictive power and robustness are improved. The randomization parameter controls not only the degree of randomness within each tree but also the degree of correlation between trees in the forest. All trees are trained independently of each other and, if possible, in parallel. Combining the predictions of all trees into a single prediction is usually done by simple averaging (*Breiman, 2001*). Thus, for the classification problem, we obtain:

$$p(c|x) = \frac{1}{T} \sum_{\tau=1}^{T} p_\tau(c|x), \tag{3}$$

where $c$ is a class, $T$ is the number of trees in the forest, $p_\tau(c|x)$ is the distribution obtained within tree $\tau$. Analogously, for the regression task, we obtain:

$$\hat{y} = \frac{1}{T} \sum_{\tau} \hat{y}_\tau, \tag{4}$$

where $\hat{y}_\tau$ is the value predicted by tree $\tau$. Alternatively to the averaging procedure, the prediction for the classification can be obtained by multiplying the predictions from different trees, namely: $p(c|x) = \frac{1}{Z} \prod_{\tau=1}^{T} p_\tau(c|x)$, where $Z$ is the normalization constant. In general, a product-based ensemble model may be less robust with respect to noise (*Criminisi, Shotton & Konukoglu, 2012*).

The construction of trees and their predictive abilities depend on model parameters. The parameters that most affect the behavior of the random forest are the following: the maximum tree depth $D$; the degree of randomization (controlled by the parameter $\rho$) and the type of randomization; the size of the forest, that is, the number of trees $T$; and the target function. A number of articles have shown how accuracy on the test sample increases monotonically with increasing forest size, $T$ (*Yin et al., 2007*; *Johnson & Shotton, 2010*; *Criminisi et al., 2010*). *Shotton et al. (2013)* showed that training very deep trees leads to overfitting. The classical work of *Breiman (2001)* also showed the importance of randomization and its effect on tree correlation. In addition, *Criminisi, Shotton & Konukoglu (2012)* determined that the randomization model directly affects the generalization ability of the random forest for the classification problem.

## APPLICATION OF PARAMETRIC ENTROPIES IN RANDOM FORESTS

### Classification task

The target function for selecting the splitting of each internal node is usually based on Shannon entropy. In this work, we consider three target functions based on the following parametric entropies: Renyi entropy, Tsallis entropy, and Sharma-Mittal entropy. Analogously to Eq. (1), Renyi entropy-based information gain for node $j$ can be formulated as follows:

$$I_j = H(S_j) - \frac{|S_j^L|}{|S_j|} H(S_j^L) - \frac{|S_j^R|}{|S_j|} H(S_j^R),$$

$$H(S_j) = \frac{1}{1-\alpha} \log \left( \sum_{c \in C} (p(c))^\alpha \right). \tag{5}$$

In Eq. (5), $H(\cdot)$ is Renyi entropy with parameter $\alpha$, $C$ is the set of classes, $c$ is a class label.
Tsallis entropy-based information gain is calculated as follows:

$$I_j = H(S_j) - \frac{|S_j^L|}{|S_j|} H(S_j^L) - \frac{|S_j^R|}{|S_j|} H(S_j^R),$$

$$H(S_j) = \frac{1}{\beta - 1} \left( 1 - \sum_{c \in C} (p(c))^\beta \right). \tag{6}$$

In equation Eq. (6), $H(\cdot)$ refers to Tsallis entropy with parameter $\beta$.

Finally, Sharma-Mittal entropy-based information gain is expressed as follows:

$$I_j = H(S_j) - \frac{|S_j^L|}{|S_j|} H(S_j^L) - \frac{|S_j^R|}{|S_j|} H(S_j^R),$$

$$H(S_j) = \frac{1}{1-\beta} \left( \left( \sum_{c \in C} (p(c))^\alpha \right)^{\frac{1-\beta}{1-\alpha}} - 1 \right). \tag{7}$$

In Eq. (7), $H(\cdot)$ is Sharma-Mittal entropy with parameters $\alpha$ and $\beta$. Let us notice that Sharma-Mittal entropy becomes Renyi entropy if $\beta \to 1$ and Tsallis entropy if $\alpha \to \beta$ (*Akturk, Bagci & Sever, 2007*). To build decision trees for the random forest, one has to maximize the chosen target function to find the best split for each internal node. Then, each leaf node makes a prediction based on the majority principle, *i.e.*, the class is chosen to which the largest number of training data points corresponds.

## Regression task

Let us briefly discuss some properties of multiple linear regression with Gaussian noise, which will then be used in the formulation of information gain. Let us consider the equation of such a regression: $y = a_0 + a_1 x_1 + \ldots + a_p x_p + \epsilon$, $p$ is the number of factors, $\epsilon \sim N(0, \sigma^2)$ and noise is independent between different measurements. Then $y|x \sim N(x^T a, \sigma^2)$, where $x = \begin{pmatrix} 1 \\ x_1 \\ \vdots \\ x_p \end{pmatrix}$ and $a = \begin{pmatrix} a_0 \\ a_1 \\ \vdots \\ a_p \end{pmatrix}$. In matrix form, it can be expressed as follows: $Y = X \cdot a + \epsilon$, where $\epsilon \sim MVN(0, \sigma^2 I)$, $X = \begin{pmatrix} 1 & x_{11} & x_{12} & \ldots & x_{1p} \\ \ldots & \ldots & \ldots & \ldots & \ldots \\ 1 & x_{n1} & x_{n2} & \ldots & x_{np} \end{pmatrix}$, $Y = \begin{pmatrix} y_1 \\ \vdots \\ y_n \end{pmatrix}$, $n$ is the number of observations.

The estimations of vector $a$ and parameter $\sigma^2$ can be obtained in the following way:

$$\hat{a} = (X^T X)^{-1} X^T Y,$$

$$\hat{\sigma}^2 = \frac{1}{n} h^T h, \text{ where } h = Y - X \cdot \hat{a}. \tag{8}$$

In the framework of a decision tree, let us define a probabilistic linear predictor (*Criminisi, Shotton & Konukoglu, 2012*) for node $j$ as $d_j(x) = x^T \hat{a}_j$, where $\hat{a}_j$ is calculated based on the observations fallen into node $j$ according to Eq. (8). Further, let us introduce four types of information gain based on classical Shannon entropy and three parametric

entropies taking into account the introduced probabilistic linear predictor. First, Shannon entropy-based information gain for node $j$ can be determined as follows:

$$I_j = H(S_j) - \frac{|S_j^L|}{|S_j|}H(S_j^L) - \frac{|S_j^R|}{|S_j|}H(S_j^R),$$

$$H(S_j) = -\frac{1}{|S_j|}\sum_{x \in S_j}\int_{y \in R}p(y|x)\log(p(y|x))dy, \qquad (9)$$

where $p(y|x) = N(x^T\hat{a}_j, \hat{\sigma}_j^2)$, $|S_j|$ is the number of observations in node $j$. In Eq. (9), $H(\cdot)$ is differential Shannon entropy. After calculating the integral, we obtain $\int_{y \in R}p(y|x)\log(p(y|x))dy = \frac{1}{2}\log(2\pi e\hat{\sigma}_j^2)$ (*Cover & Thomas, 2006*). Thus, $H(S_j) = \frac{1}{|S_j|}\sum_{x \in S_j}\frac{1}{2}\log(2\pi e\hat{\sigma}_j^2) = \frac{1}{2}\log(2\pi e\hat{\sigma}_j^2)$. Therefore, Eq. (9) can be rewritten as follows:

$$I_j = \frac{1}{2}\log(2\pi e) + \log(\hat{\sigma}_j) - \sum_{i \in \{L,R\}}\frac{|S_j^i|}{|S_j|}\left(\frac{1}{2}\log(2\pi e) + \log(\hat{\sigma}_{ji})\right). \qquad (10)$$

Second, let us introduce Renyi entropy-based information gain for node $j$:

$$I_j = H_\alpha(S_j) - \frac{|S_j^L|}{|S_j|}H_\alpha(S_j^L) - \frac{|S_j^R|}{|S_j|}H_\alpha(S_j^R),$$

$$H_\alpha(S_j) = \frac{1}{|S_j|}\sum_{x \in S_j}\frac{1}{1-\alpha}\log\left(\int_{y \in R}p^\alpha(y|x)dy\right), \qquad (11)$$

where $p(y|x) = N(x^T\hat{a}_j, \hat{\sigma}_j^2)$, $H_\alpha(\cdot)$ is differential Renyi entropy with parameter $\alpha$; more precisely, it is a conditional Renyi entropy (*Fehr & Berens, 2014*). One can demonstrate that

$$\int_{y \in R}p^\alpha(y|x)dy = \frac{1}{\hat{\sigma}^{\alpha-1}(\sqrt{2\pi})^{\alpha-1}\sqrt{\alpha}}. \qquad (12)$$

For example, this is shown in Nielsen and Nock (*Nielsen & Nock, 2011*). Thus, $H_\alpha(S_j) = \log(\sqrt{2\pi}) + \log(\hat{\sigma}_j) - \frac{\log(\sqrt{\alpha})}{1-\alpha}$. Therefore, Eq. (11) can be rewritten in the following form:

$$I_j^\alpha = \log(\sqrt{2\pi}) + \log(\hat{\sigma}_j) - \frac{\log(\sqrt{\alpha})}{1-\alpha} - \sum_{i \in \{L,R\}}\frac{|S_j^i|}{|S_j|}\left(\log(\sqrt{2\pi}) + \log(\hat{\sigma}_{ji}) - \frac{\log(\sqrt{\alpha})}{1-\alpha}\right). \qquad (13)$$

Third, Tsallis entropy-based information gain for node $j$ can be formulated in the following way:

$$I_j^\beta = H_\beta(S_j) - \frac{|S_j^L|}{|S_j|}H_\beta(S_j^L) - \frac{|S_j^R|}{|S_j|}H_\beta(S_j^R),$$

$$H_\beta(S_j) = \frac{1}{|S_j|}\sum_{x \in S_j}\frac{1}{\beta-1}\left(1 - \int_{y \in R}p^\beta(y|x)dy\right), \qquad (14)$$

where $p(y|x) = N(x^T\hat{a}_j, \hat{\sigma}_j^2)$, $H_\beta(\cdot)$ is differential Tsallis entropy with parameter $\beta$. Using relation Eq. (12), we obtain that $H_\beta(S_j) = \frac{1}{|S_j|}\sum_{x \in S_j}\frac{1}{\beta-1}\left(1 - \frac{1}{\hat{\sigma}_j^{\beta-1}(\sqrt{2\pi})^{\beta-1}\sqrt{\beta}}\right)$. Therefore,

Eq. (14) can be rewritten as:

$$I_j^{\beta} = \frac{1}{\beta-1}(1 - \frac{1}{\hat{\sigma}_j^{\beta-1}(\sqrt{2\pi})^{\beta-1}\sqrt{\beta}}) - \sum_{i\in\{L,R\}} \frac{|S_j^i|}{|S_j|}(\frac{1}{\beta-1}(1 - \frac{1}{\hat{\sigma}_{ji}^{\beta-1}(\sqrt{2\pi})^{\beta-1}\sqrt{\beta}})). \quad (15)$$

Fourth, let us introduce Sharma-Mittal entropy-based information gain for node $j$:

$$I_j = H_{\alpha,\beta}(S_j) - \frac{|S_j^L|}{|S_j|}H_{\alpha,\beta}(S_j^L) - \frac{|S_j^R|}{|S_j|}H_{\alpha,\beta}(S_j^R),$$

$$H_{\alpha,\beta}(S_j) = \frac{1}{|S_j|}\sum_{x\in S_j}\frac{1}{1-\beta}((\int_{y\in R} p^{\alpha}(y|x)dy)^{\frac{1-\beta}{1-\alpha}} - 1), \quad (16)$$

where $p(y|x) = N(x^T\hat{a}_j, \hat{\sigma}_j^2)$, $H_{\alpha,\beta}(\cdot)$ is differential Sharma-Mittal entropy with parameter $\alpha$ and $\beta$. Let us note that $\alpha, \beta > 0$, $\alpha \neq 1$, $\beta \neq 1$, $\alpha \neq \beta$ according to the definition of Sharma-Mittal entropy. Using Eq. (12), we obtain that $H_{\alpha,\beta}(S_j) = \frac{1}{|S_j|}\sum_{x\in S_j}\frac{1}{1-\beta}(\frac{1}{\hat{\sigma}_j^{\beta-1}(\sqrt{2\pi})^{\beta-1}\alpha^{\frac{1-\beta}{2(1-\alpha)}}} - 1)$. Therefore, Eq. (16) can be rewritten as follows:

$$I_j^{\alpha,\beta} = \frac{1}{1-\beta}(\frac{1}{\hat{\sigma}^{\beta-1}(\sqrt{2\pi})^{\beta-1}\alpha^{\frac{1-\beta}{2(1-\alpha)}}} - 1) -$$

$$\sum_{i\in\{L,R\}} \frac{|S_j^i|}{|S_j|}(\frac{1}{1-\beta}(\frac{1}{\hat{\sigma}_{ji}^{\beta-1}(\sqrt{2\pi})^{\beta-1}\alpha^{\frac{1-\beta}{2(1-\alpha)}}} - 1)). \quad (17)$$

In the framework of the introduced formulations of information gain, the best splitting corresponds to the maximum information gain value.

## Random forest algorithm

Let us formulate the general algorithm of random forest construction considering the proposed expressions of information gain with different types of entropies. Suppose we have the following data $(x_1, y_1), (x_2, y_2), .., (x_n, y_n)$ where each $x_i$ represents a feature vector $[x_{i_1}, x_{i_2}, \ldots, x_{i_p}]$ and let $T$ be the number of trees we want to construct in our forest. Then, to build a random forest, the following steps should be performed:

- For $t = 1, \ldots, T$:

    - Draw a bootstrap sample of size n from the data.
    - Grow a decision tree $t$ from the bootstrapped sample by repeating the following steps until the minimum node size specified beforehand is reached, or the maximum depth of the tree is reached, or the minimum reduction in the information gain is reached:

        * sample $m = \sqrt{p}$ or $m = [p/3]$ features (where p is the number of features in the dataset)
        * compute the information gain according to one of the Eqs. (5), (6), (7) for classification task and according to one of the Eqs. (10), (13), (15), (17) for regression task for each possible value among the bootstrapped data and $m$ features
        * find the information gain maximum and split the node into two children nodes

**Table 1 Summary of the datasets used for classification task.**

| Dataset | # of instances | # of features | # of classes |
|---|---|---|---|
| har | 10,299 | 561 | 6 |
| eeg-eye-state | 14,980 | 14 | 2 |
| shuttle | 58,000 | 9 | 7 |

**Table 2 Summary of the datasets used for regression task.**

| Dataset | # of instances | # of features | # of distinct values in target variable |
|---|---|---|---|
| diabetes | 442 | 10 | 214 |
| bank32nh | 8,192 | 32 | 6,284 |
| sulfur | 10,081 | 6 | 9,368 |

- Output the ensemble of trees and combine the prediction of all trees into a single prediction according to Eq. (3) for a classification task and Eq. (4) for regression.

## DESCRIPTION OF COMPUTER EXPERIMENTS

To test the proposed formulations of information gain, we used three datasets for the classification task and three datasets for the regression task. The datasets were chosen according to the following considerations: (1) Datasets should be from different areas so they reflect different types of data. Accordingly, it allows us to demonstrate that the proposed entropy approach is applicable to different types of data. (2) Datasets should be well-known in the field of machine learning. Thus, we consider the datasets that were used in other works as benchmarks. Let us note that some of the considered datasets are not balanced regarding classes. The considered datasets are summarized in Tables 1 and 2. Classification datasets are available at https://doi.org/10.5281/zenodo.8322044, and regression datasets are available at https://doi.org/10.5281/zenodo.8322236.

For the classification task, the number of trees in the forest was fixed as 300; the datasets were split into training and test sets in proportions of 0.75 and 0.25, correspondingly. Let us note that the number of trees was chosen deliberately large in order to demonstrate the advantage of our entropy approach since, according to a set of works (*Yin et al., 2007*; *Johnson & Shotton, 2010*; *Criminisi et al., 2010*; *Criminisi, Shotton & Konukoglu, 2012*), increasing the number of trees leads to a monotonous increase in the accuracy of the model and a decrease in fluctuations of the results. The number of features sampled for each split of a node in decision trees was set to $[p/3]$, where $p$ is the number of features in the dataset. The maximum depth of decision trees was set to 16. In addition, the learning of each decision tree was organized not on bootstrap samples but on the whole dataset. Otherwise, the random forest was constructed as described in the 'Random forest algorithm' section. Parameters $\alpha$ and $\beta$ were varied in the range of $[0.01, \ldots, 0.99]$ in the increments of 0.01. To estimate the quality of random forest predictions, accuracy, and F1 score were calculated for the test sets, and a baseline model of random forest with Shannon entropy-based information gain was calculated. Let us note that we did not consider ROC and AUC

metrics since it is preferable to use F1 score for imbalanced datasets, while optimization with ROC and AUC is not optimal in this case.

For the regression task, the number of trees in the forest was fixed as 500; the datasets were split into training and test sets in proportions of 0.75 and 0.25. The number of features sampled for each split of a node in decision trees was set to $\sqrt{p}$, where $p$ is the number of features in the dataset. The maximum depth of decision trees was set to 16. The minimum number of observations in a node was set as $p \cdot 10$ to have sufficient data for regression training (the so-called rule of 10). The random forest was constructed as described in the 'Random forest algorithm' section. To estimate the quality of random forest predictions, $R^2$, MSE, and MAE scores were calculated for the test sets. For diabetes dataset, parameters $\alpha$ and $\beta$ were varied in the range of $[0.01, \ldots, 0.99]$ in the increments of 0.01. However, for relatively large datasets (bank32nh and sulfur), parameters $\alpha$ and $\beta$ were varied in the range of $[0.1, \ldots, 0.9]$ in the increments of 0.1 (small grid of parameter values) for calculations with Sharma-Mittal entropy-based information gain in order to reduce computational and time costs. For Renyi and Tsallis entropies, parameters $\alpha$ and $\beta$ were varied in the range of $[0.01, \ldots, 0.99]$ in the increments of 0.01.

To estimate the proposed models, we consider two baseline models, namely, Breiman's random forest for regression with a least-square mean target function implemented in sklearn and simple multiple linear regression, also implemented in sklearn (https://scikit-learn.org/0.21/, scikit-learn version 0.21.2) (*Pedregosa et al., 2011*). Let us note that the configuration of Breiman's random forest was also set as described above, *i.e.,* 500 trees with a maximum depth of 16, and $\sqrt{p}$ sampled features for each split.

The source codes of the random forest algorithms with the proposed information gains can be found at https://doi.org/10.5281/zenodo.8327384.

## RESULTS

Figure 1 demonstrates the dependence of accuracy for the classification task on the used entropy type and its parameter value for each dataset. Figure 1A demonstrates the results achieved with the application of Renyi entropy while Fig. 1B demonstrates the behavior of accuracy when Tsallis entropy is used. One can see that the accuracy increases with the increasing parameter for both Renyi and Tsallis entropies and for all three datasets. Moreover, the lines of accuracy corresponding to Renyi and Tsallis entropies are above the accuracy line of the baseline model with Shannon entropy for all three datasets, meaning that the application of deformed entropies improves the quality of predictions for random forest algorithms. The best accuracy values for different types of information gain and the accuracy value of the baseline model with Shannon entropy-based information gain (*i.e.,* Breiman's algorithm) are presented in Table 3 for each dataset. The best accuracy values for each dataset are highlighted in bold. One can see that applying two-parametric Sharma-Mittal entropy allows us to improve the prediction accuracy further, demonstrating the best-achieved results (Table 3) for all three datasets.

Figure 2 demonstrates the dependence of $R^2$ value on entropy type and its parameter value for each dataset for the regression task. One can see that $R^2$ does not have a clear

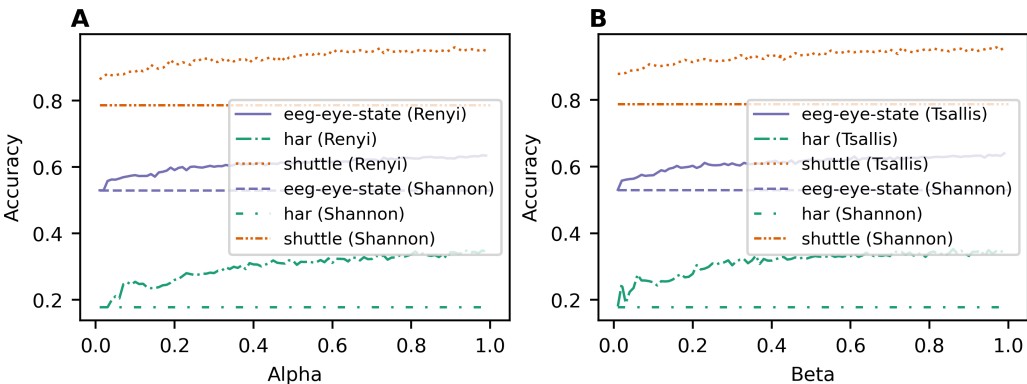

**Figure 1** Dependence of accuracy on entropy type and corresponding parameter (classification task). (A) Renyi entropy-based information gain. (B) Tsallis entropy-based information gain.

**Table 3** Accuracy of random forest algorithms for classification with different types of information gain.

| Dataset | Renyi | Tsallis | Sharma-Mittal | Shannon (baseline) |
|---|---|---|---|---|
| har | 0.349 ($\alpha = 0.98$) | 0.353 ($\beta = 0.86$) | **0.377** ($\alpha = 0.89, \beta = 0.11$) | 0.178 |
| eeg-eye-state | 0.635 ($\alpha = 0.98$) | 0.639 ($\beta = 0.99$) | **0.649** ($\alpha = 0.99, \beta = 0.05$) | 0.53 |
| shuttle | 0.9607 ($\alpha = 0.91$) | 0.9576 ($\beta = 0.97$) | **0.9612** ($\alpha = 0.94, \beta = 0.92$) | 0.787 |

monotone behavior as it was for the accuracy for the classification task. Let us note that the introduction of parameterized entropies in the random forest algorithm differs greatly for classification and regression versions. This is why the behavior of the obtained results for classification and regression are so different. Indeed, $R^2$ values for random forests with Renyi and Tsallis entropy-based information gains fluctuate around a certain value, which is specific for each dataset, in the range of $\pm 0.006$. This result means that the proposed models are rather stable under variation of hyperparameters of the parameterized entropies. Moreover, one can see that the lines of $R^2$ for entropy-based information gains lie above the $R^2$ values of baseline models for all three datasets, meaning that the application of an entropy-based information gain improves the quality of predictions for random forest algorithms. The best values of $R^2$, MSE, and MAE for different types of information gain for each dataset and corresponding values for baseline models are given in Table 4. The best values for each dataset are highlighted in bold.

## DISCUSSION

First of all, let us note that the quality of obtained results for the proposed information gains (Tables 3 and 4) can be significantly improved by varying the number of sampled features, the minimum number of samples in a leaf node and increasing the number of trees. However, such tuning depends on a dataset and is time-consuming. Therefore, in this work, we present the results without such a tuning.

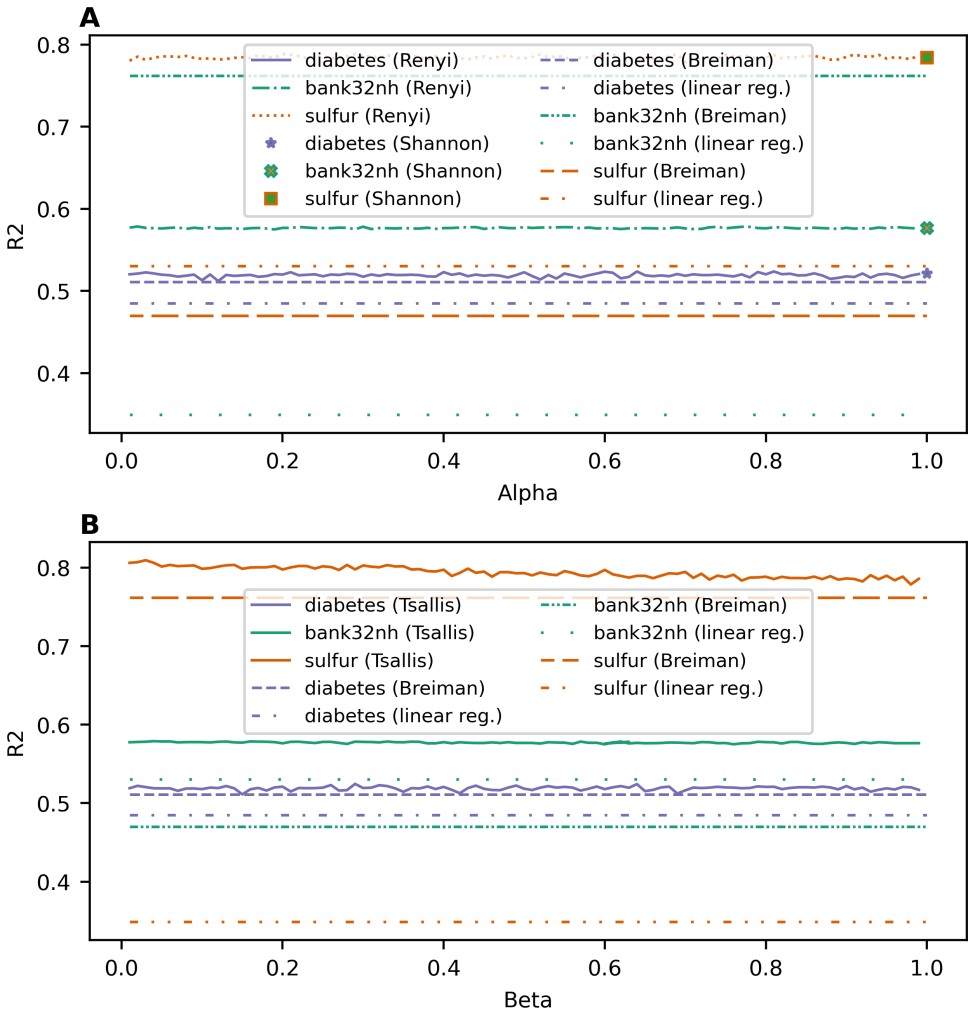

**Figure 2** Dependence of $R^2$ on entropy type and corresponding parameter (regression task). (A) Renyi entropy-based information gain. (B) Tsallis entropy-based information gain.

Second, based on our results, the application of deformed entropies allows us to increase the quality of predictions compared to baseline models for both classification and regression tasks and for all tested datasets. Let us note that, for the classification task, the accuracy of predictions when using deformed entropies is higher for all considered entropy parameter values compared to the accuracy of the algorithm with classical Shannon-based information gain. For the regression task, the values of $R^2$ for the proposed types of entropy-based information gain are larger than that for standard algorithms, such as Breiman's random forest and multiple linear regression.

Third, we discovered that application of two-parametric Sharma-Mittal entropy had the best results in terms of prediction accuracy for the classification task. For the regression task for diabetes dataset, where a large grid of parameter values was considered, the best results were also obtained with Sharma-Mittal entropy-based information gain. For larger

**Table 4  Results on prediction quality of random forest algorithms with different types of information gain for a regression problem.**

| Dataset | Measure | Shannon | Renyi | Tsallis | Sharma-Mittal | Breiman's RF (baseline) | Linear regression (baseline) |
|---------|---------|---------|-------|---------|---------------|------------------------|------------------------------|
| diabetes | $R^2$ | 0.521487 | 0.52366 ($\alpha = 0.64$) | 0.5245 ($\beta = 0.29$) | **0.5265** ($\alpha = 0.18, \beta = 0.61$) | 0.51097 | 0.4849 |
| | MSE | 2646.026 | 2633.997 ($\alpha = 0.64$) | 2629.324 ($\beta = 0.29$) | **2618.3** ($\alpha = 0.18, \beta = 0.61$) | 2704.178 | 2848.295 |
| | MAE | 39.664 | 39.448 ($\alpha = 0.79$) | 39.477 ($\beta = 0.29$) | **39.322** ($\alpha = 0.97, \beta = 0.07$) | 41.934 | 41.548 |
| bank32nh | $R^2$ | 0.57653 | 0.5784 ($\alpha = 0.02$) | **0.57893** ($\beta = 0.04$) | 0.57885 ($\alpha = 0.5, \beta = 0.5$) | 0.469 | 0.53 |
| | MSE | 0.00623 | 0.006204 ($\alpha = 0.02$) | **0.006197** ($\beta = 0.04$) | 0.006198 ($\alpha = 0.5, \beta = 0.5$) | 0.008 | 0.007 |
| | MAE | 0.05285 | **0.05279** ($\alpha = 0.42$) | 0.052825 ($\beta = 0.57$) | 0.052806 ($\alpha = 0.3, \beta = 0.6$) | 0.061 | 0.058 |
| sulfur | $R^2$ | 0.7844 | 0.7894 ($\alpha = 0.61$) | **0.8094** ($\beta = 0.03$) | 0.8085 ($\alpha = 0.2, \beta = 0.1$) | 0.762 | 0.349 |
| | MSE | 0.00063 | 0.00062 ($\alpha = 0.61$) | **0.00056** ($\beta = 0.03$) | 0.000563 ($\alpha = 0.2, \beta = 0.1$) | 0.0007 | 0.0019 |
| | MAE | 0.01059 | 0.01054 ($\alpha = 0.17$) | 0.010478 ($\beta = 0.4$) | **0.010475** ($\alpha = 0.2, \beta = 0.1$) | 0.0106 | 0.0189 |

datasets, where a small grid of parameter values was considered for experiments with Sharma-Mittal entropy, the best results in terms of $R^2$ are obtained with Tsallis entropy.

Speaking of the advantages of our approach, we can determine the following. The proposed modifications of random forest algorithms based on parameterized entropies for classification and regression problems have a significant degree of flexibility, allowing us to qualitatively determine patterns inherent in different types of data. First, flexibility is due to the fact that non-classical entropies better describe different types of distributions (for example, power law distribution), which differ significantly from the normal distribution. Second, parameterized entropies provide additional parameter settings, which allow us to fit trees on specific feature sets more accurately. That is why combining these two factors makes it possible to get better results. However, on the other hand, the presence of additional parameter settings leads to a significant increase in the computation time. This limitation can be solved by parallelizing the computations using CUDA technology; however, this is beyond the scope of this work. Speaking of other possible limitations or disadvantages of this work, it should be noted that our approach is tested only on datasets with numeric features, and datasets with categorical or one-hot encoded features are not considered. Another possible disadvantage of our work is that the proposed algorithms in Python are not well optimized and, therefore, are more time-consuming than baseline models. In addition, the proposed algorithm for regression forest is more computationally expensive (namely, because for each split, a set of linear regression models is computed) than classical Breiman's random forest.

# CONCLUSION

In this work, we proposed a generalized version of the random forest algorithm, which uses parameterized entropies of Renyi, Tsallis, and Sharma-Mittal for classification and regression problems. Generalization is achieved due to the fact that the classical version of the random forest algorithm for classification problems is based on Shannon entropy, which is a special case of Renyi and Tsallis entropies, which, in turn, are special cases of the two-parameter Sharma-Mittal entropy. The use of additional parameters in entropies

makes it easy to take into account specialized patterns in complex data that are difficult to determine when using the classical version of the random forest algorithm. New variants of the random forest algorithm were tested on six different datasets (three for the classification problem and three for the regression problem). As our experimental results demonstrate, parametrized entropies improve classification results practically by 19–111% (in terms of accuracy metric) depending on the dataset type. In addition, in regression problems, our approach shows an improvement in the range of 2–23% (depending on the dataset) in terms of the quality measure $R^2$. Thus, our entropy approach to classification and regression random forests is novel and demonstrates a better result compared to the classical approach when applied to complex data analysis.

## ACKNOWLEDGEMENTS

This research made use of computational resources of the HPC facilities at HSE University.

### Funding

This work was supported by the Basic Research Program at the National Research University Higher School of Economics in 2023 (project "Innovative methods of data collection and analysis in the modeling of communicative behavior of Internet users and the development of respective technological solutions"). The funders had no role in study design, data collection and analysis, decision to publish, or preparation of the manuscript.

### Grant Disclosures

The following grant information was disclosed by the authors:
The Basic Research Program at the National Research University Higher School of Economics in 2023.

### Competing Interests

The authors declare there are no competing interests.

### Author Contributions

- Vera Ignatenko conceived and designed the experiments, performed the experiments, analyzed the data, performed the computation work, prepared figures and/or tables, authored or reviewed drafts of the article, and approved the final draft.
- Anton Surkov performed the experiments, performed the computation work, authored or reviewed drafts of the article, and approved the final draft.
- Sergei Koltcov conceived and designed the experiments, analyzed the data, authored or reviewed drafts of the article, and approved the final draft.

### Data Availability

    The source code are available at Zenodo: veraignatenko. (2023). hse-scila/random_forest_project: For Zenodo (v1.0.0). Zenodo. https://doi.org/10.5281/zenodo.8327384.

The classification datasets are available at Zenodo: Ignatenko Vera. (2023). Classification datasets [Data set]. Zenodo. https://doi.org/10.5281/zenodo.8322044.

The regression datasets are available at Zenodo: Ignatenko Vera. (2023). Regression datasets [Data set]. Zenodo. https://doi.org/10.5281/zenodo.8322236.

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
