# Peer review of "Random forests with parametric entropy-based information gains for classification and regression problems"

_PeerJ Computer Science, doi:10.7717/peerj-cs.1775_

## Round 0.1 · original submission · Major Revisions

Dear authors,

Your article has not been recommended for publication in its current form. However, we do encourage you to address the concerns and criticisms of the reviewers especially on readability and quality and resubmit your article once you have updated it accordingly.

Best wishes,

**Language Note:** The review process has identified that the English language must be improved. PeerJ can provide language editing services - please contact us at [email protected] for pricing (be sure to provide your manuscript number and title). Alternatively, you should make your own arrangements to improve the language quality and provide details in your response letter. – PeerJ Staff

·

Basic reporting

1) In the Intro section " it is known that complex data are better described with nonclassical
50 deformed entropies" Please explain this statement along with reference

2) Try to explain the Basics of Random Forest and APPLICATION OF PARAMETRIC ENTROPIES IN RANDOM FORESTS section in simple words also. So that non technical audience can also understand and relate well

3) Make sure the sources mentioned in Table 1 and Table 2 are in right format

4) Make sure the sources mentioned for sklearn are in right format. Please refer to PeerJ author guide for the same

5) In Figure 1 and Figure 2, the color labels are missing

6) Table 4 only contains data for Regression model, why not classification model ?

Experimental design

1) In experiments section "The maximum 220 depth of decision trees was set to 16. The minimum number of observations in a node was set as p =10" Any reason for why p=10 ? Please explain

2) explaining a bit more about the data like prediction classes, are those classes balanced or unbalanced? and if unbalanced how are they balanced ?

Validity of the findings

1) for the findings and performance of the model only accuracy and F1 were used why not AUC and ROC metrics ?

2) in the results section " One can see that R does not have a clear monotone behavior as it was for the accuracy for the classification task" .. Please explain why ? why not ?

2) in the results section " One can see that R does not have a clear monotone behavior as it was for the accuracy for the classification task" .. Please explain why ? why not ?

3) in the statement "R2 values for random forests with Renyi and Tsallis entropy-based information gains fluctuate around a certain value, which is specific for each dataset, in the range of ±0.006 " Please explain why ? why not ?

·

Basic reporting

The English language used in the study is clear and sufficient. However, there are spell mistakes in some places.

The sources given in the study are sufficient and up-to-date. Also, tables and figures are self explanatory. However, before the Results section, a graphical abstract of the study should be given and the abstract should be explained.

The abstract should be digitized. At the end of the abstract, the numerical values of the achievements should be given.

At the end of the Introduction, a paragraph should be opened before the last paragraph and the highlights of the study should be emphasized.

The Conclusion should be enhanced.

Experimental design

On what basis were the datasets selected for the study chosen? Does it have a specific purpose or does it involve randomly chosen structures?

Why is the number of trees in the random forest set to 300? However, the data is divided into two as training and testing. Does cross-validation affect results? Can it affect positively or negatively? These should be discussed.

Validity of the findings

The results given in both Table 3 and Table 4 clearly show that the proposed method is effective. However, today there are a large number of residual classification models. There are different approaches in the field of machine learning, especially deep learning models. Analysis operations with these models can produce better results than the random forest. It will be more informative about the random forest, which is the focus of the study, if the authors clearly state why they chose the random forest and the advantages of this model over other models.

The advantages and disadvantages of the study should be added to the discussion section and its contributions to the literature should be mentioned.

Additional comments

In the study, the researchers aimed to increase the performance of the random forest classifier and emphasized that the performance of the classifier could be increased by using parametric entropies instead of the traditional Shannon entropy. Although the study is an interesting study, eliminating the issues will increase the readability and quality of the study.

---

## Round 0.2 · accepted · Accept

Dear authors,

We have received the reviewers' reports on your manuscript. Thank you for addressing the concerns, questions and suggestions on the review reports. The paper seems to have improved in the opinion of the reviewers. At the moment your article seems to be acceptable for publication after the last revision.

Best wishes,

·

Basic reporting

Basic reporting
Based on comments from previous review, the author has answer all the question thoroughly and covered the missing gap between the Literature review.


I suggest the authors consider adding a supplementary appendix with:

Formal definitions for all parameterized entropies used (Renyi, Tsallis, Sharma-Mittal)
Detailed derivations for the information gain formulas proposed
Precise statements of any theorems/lemmas on properties of the information gains

Experimental design

Based on comments from previous review, the author has answer all the question thoroughly

Please mention
Are there additional technical details that could be provided to facilitate reproducibility by other researchers?
Did you identify any specific ethical considerations that should be more explicitly addressed?

Validity of the findings

Based on comments from previous review, the author has answer all the question thoroughly

Please explain based on findings, What is the potential real-world impact of this work if adopted? How significantly could it improve random forest performance in applications? (if not)

Additional comments

Article is written in well English and shows the problem really well.

·

Basic reporting

The researchers carried out the revision.

Experimental design

The researchers carried out the revision.

Validity of the findings

The researchers carried out the revision.

Additional comments

The researchers carried out the revision.